# A Critical Lens on Health: Key Principles of Critical Discourse Analysis and Its Benefits to Anti-Racism in Population Public Health Research

Jessica Naidu [1] , Elizabeth Oddone Paolucci [1] and Tanvir C. Turin [1,2,*]

1   Department of Community Health Sciences, Cumming School of Medicine, University of Calgary, Calgary, AB T2N 4N1, Canada
2   Department of Family Medicine, Cumming School of Medicine, University of Calgary, Calgary, AB T2N 4N1, Canada
*   Correspondence: turin.chowdhury@ucalgary.ca

**Abstract:** Critical discourse analysis (CDA) is an interdisciplinary research methodology used to analyze discourse as a form of "social practice", exploring how meaning is socially constructed. In addition, the methodology draws from the field of critical studies, in which research places deliberate focus on the social and political forces that produce social phenomena as a means to challenge and change societal practices. The purpose of this article is to demonstrate the benefits of CDA to population public health (PPH) research. We will do this by providing a brief overview of CDA and its history and purpose in research and then identifying and discussing three crucial principles that we argue are crucial to successful CDA research: (1) CDA research should contribute to social justice; (2) CDA is strongly based in theory; and (3) CDA draws from constructivist epistemology. A key benefit that CDA brings to PPH research is its critical lens, which aligns with the fundamental goals of PPH including addressing the social determinants of health and reducing health inequities. Our analysis demonstrates the need for researchers in population public health to strongly consider critical discourse analysis as an approach to understanding the social determinants of health and eliminating health inequities in order to achieve health and wellness for all.

**Keywords:** social justice; discourse analysis; public health research

## 1. Introduction

The term 'discourse' is used to refer to all forms of written and spoken language. Discourse analysis (DA) is a research methodology derived from the study of linguistics that analyzes the formal aspects of discourse, including basic units of speech and linguistic structures. Unlike traditional linguistics, DA goes beyond the literal meaning of what is spoken or written to explain how it operates within a social context. Thus, it analyzes how meaning is constructed through language within the context of the social world [1].

Critical discourse analysis (CDA) is an interdisciplinary research methodology used to analyze discourse. CDA, like DA, views discourse as a form of "social practice" ([2] p. 1) and investigates the social construction of meaning. In addition, the methodology draws from the field of critical studies, which deliberately focuses on the social and political forces that produce social phenomena as a means to challenge and change societal practices. Unlike DA, CDA seeks to critique and alter language usage in social practice, as opposed to merely explaining it [2].

CDA has been used as a strategy of inquiry in various disciplines, including sociology, communications studies, and psychology; it has also been used increasingly in population and public health (PPH) research [3]. The objective of PPH research is to investigate ways to prevent disease and promote health in populations [4]. Identifying the social determinants of health and reducing health inequities are integral goals of PPH research.

The social, economic, and political variables that shape health are social determinants of health (SDOH). Health inequities are unfair and avoidable disparities in health outcomes across populations [5]. Given the social and political orientation of PPH, the critical lens of CDA may be an effective study method for PPH research.

In this paper, we will present a brief overview of CDA and its research history and purpose. Then, we will define and discuss three fundamental principles that, we argue, are essential in successful CDA research, particularly in PPH: (1) CDA research should contribute to social justice; (2) CDA is strongly based in theory; and (3) CDA employs constructivist epistemology. We conclude with a critical appraisal of the methodology, focusing on its merits and limitations as well as its benefits for PPH research.

## 2. Critical Discourse Analysis Overview

### 2.1. History

As noted in the Introduction, CDA is an interdisciplinary research methodology used to analyze discourse. Historically, CDA has been synonymous with critical linguistics (CL) and critical discourse studies (CDSs). Critical linguistics can be traced back to the work of Frankfurt School social theorists from the early- to mid-twentieth century. This school was predominantly concerned with identifying and challenging socioeconomic injustices of the time [2]. For instance, Jurgen Habermas argued that "language is also a medium of social domination and force. It serves to legitimize relations of organized power" ([2] p. 2).

The term 'critical linguistics' has largely been replaced by the term 'critical discourse analysis', which can be traced back to a January 1991 symposium in Amsterdam, where a group of scholars, Teun van Dijk, Norman Fairclough, Gunther Kress, Theo van Leewen, and Ruth Wodak, convened to discuss theories and methods of linguistic research [2]. Finally, 'critical discourse studies' is a third, often interchangeable term, denoting a broader scope and application of the method [6]. "Critical" is the central notion in each of these interchangeable terms. In this context, to be critical means to notice and oppose the ways in which discourse is used to socially construct truth and enforce power and control [1]. For the purposes of this paper, we will use the term critical discourse analysis (CDA). Presented subsequently is a summary of the CDA's most important tenets.

### 2.2. The Critical Impetus

In the spirit of what Wodak and Meyer refer to as the Critical Impetus, CDA scholars focus on critiquing and changing society rather than merely describing and explaining it. Here, critical research ought to "be directed at the totality of society in its historical specificity", which means that it ought to be contextualized within the social, political, cultural, and historical spheres. Critical research must also be interdisciplinary, "improving the understanding of society by integrating all the major social sciences, including economics, sociology, history, political science, anthropology and psychology" ([6] p. 7). Critical research, including CDA, aims to produce knowledge that enables individuals to liberate themselves or others from forms of dominance and discrimination [6]; thus, this impetus is in the spirit of eradicating social injustice.

### 2.3. Michel Foucault's Theory of Power

In addition to the aforementioned scholars, Michel Foucault has had a major influence on CDA, notably with his work on power, a core concept in CDA. Knowledge is intrinsically related to Foucauldian power. In fact, he uses the term "power/knowledge" to represent this relationship and his thesis that power is constructed by dominant forces of society through knowledge. Conversely, power is necessary for the construction of knowledge and truth [7]. Foucault notes that power is not necessarily coercive and repressive, adding that "if power were nothing but repressive . . . do you think one would be brought to obey it?" ([7] p. 119). Instead, according to Foucault, power "traverses and produces things, it induces pleasure, forms knowledge, produces discourse" ([7] p. 119). Thus, it frequently functions more surreptitiously than coercive power. It may go undetected and

unchallenged by those upon whom it imposes, and those who wield and profit from it may do so unwittingly. Recognizing the role of power is a crucial initial step in addressing power disparities.

*2.4. Ideology*

CDA recognizes that discourse is intrinsically ideological, as it is defined as "social forms and processes within which, and by means of which, symbolic forms circulate in the social world" ([2] p. 10). CDA researchers view ideology as fundamental to the establishment and maintenance of unequal power relations and strive to "demystify discourses by deciphering ideologies" ([2] p. 10) underlying them. Throughout the entirety of the research process, CDA researchers must also explicitly consider their own ideologies. This appears to contradict the objectivity often sought in research, where the scientific and the ideological are considered mutually exclusive. CDA research asserts that all research is ideological; therefore, ignoring the role of ideology in activities such as formulating a research question, collecting data, and analyzing findings is to neglect a fundamental part of what shapes a researcher's conclusions or truth claims [1].

It is important to note that these three tenets are not an exhaustive list, but rather the most relevant for the purpose of this study. In addition, it is essential to highlight that while we have separated things for the sake of description, they are interlinked. For instance, the critical impetus of CDA is to reveal ideologies and power dynamics in language. Moreover, ideologies and discourses are only likely to become dominant if the public perceives them as neutral or moderate. Thus, power in the Foucauldian sense is necessary for the imposition of an ideological standpoint as a value-free truth, as opposed to an extremist or fringe belief.

## 3. Principles for Successful Critical Discourse Analysis

In this section, we suggest that there are three essential principles for conducting effective CDA research. As opposed to instructions or suggestions on how to conduct specific activities such as data collection and analysis, these are the principles and perspectives by which CDA researchers should work. As demonstrated in the preceding discussion and as observed by many CDA scholars, there is no right or wrong way to conduct research in CDA; nonetheless, there are right and wrong ways to think and act as a CDA researcher. In the following section, we will explain how.

*3.1. CDA Research Should Contribute to Social Justice*

The first essential principle for conducting effective CDA research in population health is that CDA research should advance social justice. The objective of social justice scholars and activists is "the fair distribution of society's benefits, responsibilities and their consequences" ([8] p. 1). There is a focus on the "relative position of one social group in relationship to others in society as well as on the root causes of disparities and what can be done to eliminate them" ([8] p. 1). Thus, recognizing social power dynamics is crucial for social justice aims. This principle is intertwined with the three aforementioned tenets of critical discourse analysis. CDA is inherently critical, concerned with ideology, and is committed to exposing the power dynamics underlying the phenomena it studies in order to eliminate disparities. It is therefore closely related to the aims of social justice to achieve a fair distribution of the benefits, responsibilities, and consequences of society.

In the context of population public health, social justice is the view that everyone deserves equal rights and opportunities for good health [8]. This concept is closely related to the concept of health equity, which is a core value of population public health. Health equity refers to social justice regarding health and the opportunity to attain health. Health inequities are avoidable and unfair disparities in health outcomes across populations. These are produced and reproduced by institutions, policies, and practices that create an unequal distribution of power and resources among communities based on race, class, gender, location, and other factors. Health inequities are social injustices in health. Consequently, the eradication of health inequity entails the eradication of social injustice in health [9].

The following are examples of how this might be accomplished in a CDA research project. A CDA of how perpetrators of mass violence are discursively constructed in North American news media after 9/11 must consider the association between the perpetrator's race and whether they are discursively constructed as a terrorist or a gunman. A CDA of American drug policy should examine the construction of crack cocaine consumption among black Americans and opioid use among white Americans. An additional comparison of drug-related incarceration rates by race would be a useful component of such an analysis. Lastly, a CDA of universal health promotion messages emphasizing the importance of physical activity for health must explore how such messages further marginalize individuals with disabilities in inaccessible built environments. By identifying stereotypes in the construction of marginalized communities, and in the latter case, the construction of health in a way that further excludes a disadvantaged community, these examples illustrate research that contributes to the goal of social justice to achieve a fair distribution of society's benefits.

### 3.2. CDA Is Strongly Based in Theory

The second essential principle of effective CDA research in population public health is that CDA is theoretically grounded. CDA research requires the application of theory, typically social theory, which describes the structures and functions of society. In addition, CDA researchers must be able to adapt their theoretical claims to the tools and methods of analysis they use. Wodak and Meyer identified several key theoretical influences to consider when conducting CDA. Due to their relevance to PPH research objectives, we focus on three theoretical influences: (a) general social theories; (b) micro-sociological theories; and (c) socio-psychological theories [6].

### 3.2.1. General Social Theories

According to Wodak and Meyer, general social theories are grand theories that aim to explain the relationship between structure and the individual [6]. A noteworthy example of this type of theory is Anthony Giddens' theory of structuration. Giddens' theory integrates macro and micro sociological theories, or theories of structure and theories of agency, to explain societal processes and the formation of systems. Giddens posits that there is a "duality of structure" ([10] p. 16) in which structures and agents of society function as two inseparable sides of the same coin. As social acts are produced and reproduced throughout space and time within structures, they transform into systems. On one side of the coin, structures facilitate and restrict individual social action, thus legitimizing some social interactions and behaviors while delegitimizing others [10]. These theories are relevant for examining systemic barriers and facilitators to populations achieving optimal health and wellness in the setting of PPH.

### 3.2.2. Micro-Sociological Theories

Micro-sociological theories are those that aim to explain interactions between individuals and groups and propose that societal processes result from these human interactions [6]. These theories tend to favour highly interpretivist analyses, such as those grounded in hermeneutics. This is exemplified by symbolic interactionism (SI). This sociological theory posits that an individual's behaviour toward others is predicated on the meanings they have constructed about these persons [10]. These meanings are derived from individuals' social interactions with other individuals and society. Symbolic interactionism posits that a physical reality exists only through a person's social understanding of that reality. Thus, when people act in relation to their surroundings, they do not do so directly, but rather indirectly through the lens of their social understanding [11,12]. There are four main principles of SI. First, individuals act according to their social understanding of "objects" in their environment. For example, a person who views the "object" of the family as relatively unimportant will de-emphasize the importance of family in their decisions and actions. Second, interactions occur in a social and cultural context in which objects, people, and

situations must be defined and characterized based on an individual's social understanding. Third, social understanding is created through interactions with other individuals and society. Fourth, these social understandings are created and recreated through a process of interpretation that occurs each time a person interacts with others [13].

### 3.2.3. Socio-Psychological Theories

Socio-psychological theories focus on "the social conditions of emotions and cognition" ([6] p. 24) and, similarly to micro-sociological theories, seek to explain interactions between individuals and communities. In contrast to micro-sociological theories, these theories tend to favour causal explanations over interpretive explanations [6]. Thus, these theories may be conducive to PPH research on the reasons for behaviour modification. The health belief model is an example of this type of theory, as it is frequently used to explain causal factors that predict people's engagement in health behaviours. Perceived sensitivity to a certain health problem, perceived benefits of engaging in certain health behaviours, and perceived barriers to engaging in certain health behaviours are examples of some of these characteristics [14].

### *3.3. CDA Draws from Constructivist Epistemology*

The third essential principle of effective CDA in population public health research is that CDA draws from constructivist epistemology, which is vastly distinct from the prevalent positivist epistemological stance, frequently assumed in health science research. Constructivist research in PPH shares two characteristics: an explicit research paradigm and explicit reflexivity. We will elaborate upon these below.

### 3.3.1. Explicit Research Paradigm

The elements of a research paradigm are ontology, epistemology, methodology, and methods. For some time, ontology and epistemology have been the core of humanities and social science research. In health sciences, the idea that varying ontologies and epistemologies inform and justify the knowledge produced by research has increasingly gained traction [15]. Table 1 identifies these elements and presents examples from positivist and constructivist perspectives. These are not the only two standpoints, but they are the best at demonstrating which paradigms most adequately justify CDA research.

**Table 1.** Critical discourse analysis research paradigm.

| | |
|---|---|
| Ontology | Concerns the nature of reality/truth—"what is true?"<br>Positivist: truth is objective, single/fixed, independent of human perception, discovered/discoverable<br>Constructivist: truth is subjective, there are multiple truths, dependent on human perception across space and time, truth is socially constructed |
| Epistemology | Concerns the nature of knowledge—"how do we know what is true?"<br>Positivist: objective, non-ideological, findings are truth<br>constructivist: subjective, ideological, findings are constructed meanings |
| Methodology | Strategies of inquiry to seek truth<br>Positivist: deductive, quantitative, focus on measurement of data<br>Constructivist: inductive, qualitative, focus on interpretation of data |
| Methods | Actual activities, instruments, techniques:<br>Positivist: physical measurement, surveys, statistical analysis, structured interviews, content analysis<br>Constructivist: focus groups, unstructured interviews, semi-structured interview, discourse analysis |

The purpose of Table 1 is to explain how knowledge and truth are socially constructed through language, demonstrating CDA's constructivist orientation. Consequently, an effective CDA researcher would likely employ a constructivist research paradigm. Moreover, it

is argued that performing CDA well requires explicitly identifying one's research paradigm in their product (i.e., manuscript or presentation).

### 3.3.2. Explicit Reflexivity

The second important factor related to the constructivist principle is that effective CDA requires leveraging reflexivity. In the context of research, reflexivity is when a researcher is aware not just of the social context of their participants, but also of their own, and how their own social context influences their conduct, interpretations, and representations of data [16]. As with ontological and epistemological claims, researchers must not only keep reflexivity in the back of their minds, but also document it as a core element of their research findings. This is crucial because of the value CDA places on the connection between power and knowledge. Given that the majority of researchers are affiliated with universities, they occupy a position of social power in society. When analyzing text concerning a marginalized community, a researcher must be aware of how their position and the power that comes with it influences their conclusions. This reflexivity is necessary independent of a researcher's relative power in society, because researchers hold positions of power relative to participants in the research setting. It is especially important in projects targeting marginalized communities, who frequently hold little to no relative influence in society and the research setting. For effective and socially just research, it is necessary to consider the role of power relations in the process of knowledge construction in CDA research [1].

### 4. Critical Discourse Analysis of a Population Public Health Issue—Example

One example of critical discourse analysis of a PPH issue was performed by Reitmanova, Gustafson, and Ahmed's (2015) analysis of the Canadian Press and its implications for public health policies [17]. Using framing as a theory of media effects, the authors conducted a critical discourse analysis of 273 articles from 10 major Canadian news sources over ten years. Framing aims to explain how news media cover, construct, and represent certain stories. This requires analyzing news reports for "the presence or absence of certain keywords, stock phrases, stereotyped images, sources of information and sentences that provide thematically reinforcing clusters of facts or judgments" ([17] p. 3).

This analysis is valuable to PPH because the media influences public opinion and perception of health issues, as well as public health policy and healthcare practice. Thus, it is essential to understand how the media constructs and reports on health. More specific to this paper and to the goals of PPH, it is important to understand how the media constructs and portrays a population of Canadians who use the health system.

Reitmanova, Gustafson, and Ahmed found that Canadian news media discourses construct the immigrant body as both a disease breeder and an irresponsible fraudster [17]. Moreover, these constructs are predicated on the racialization of immigrants and immigrant health issues. The results of this study suggest: (1) the de-racialization of immigrant bodies and immigrant health issues is required for more fair and accurate media coverage on immigrant health; and (2) the transformation of the Canadian press toward greater inclusiveness. These steps are needed to create the necessary shift for immigrants to receive equitable health care access [17].

### 5. Critical Evaluation and Benefits to Population Public Health Research

*5.1. Strengths*

The main strength of CDA for PPH research is its linkage with social justice. As demonstrated thus far in this paper, CDA is a critical methodology that aims to identify and dismantle disproportionate power relations in society. Although there may be other qualitative research methodologies with links to social justice, CDA has been demonstrated to align with social justice and the core competencies of PPH. According to Edwards and Davison, PPH uses advocacy, policy change, and social interventions to improve collective health; thus, social justice is a core value of PPH [8]. This is reflected in the Public Health Agency of Canada's core competencies. The competencies that align with social justice and

CDA principles are shown in Table 2, adapted from Edwards and Davison (2008). In light of this, we contend that CDA is a critical qualitative method that is ideal for population public health research.

**Table 2.** PHAC core competencies, social justice, and CDA alignment.

| Domain of PPH Core Competencies | Alignment with Social Justice | Alignment with CDA |
|---|---|---|
| Public Health Sciences | Understand relationships between social determinants of health and inequities | Critical impetus Social justice Ideology |
| Assessment and Analysis | Work with marginalized populations to use data to examine and act on health inequities | Critical impetus Social justice Ideology |
| Diversity and Inclusiveness | Understand and apply the Universal Declaration on Human Rights | Critical impetus Social justice Ideology constructivism |
| Communication | Develop communication strategies for subpopulations that have been historically oppressed | Critical impetus Social justice Ideology constructivism |

PPH research often disregards the ideological dimension of health. As noted by Lupton (1992), public health professionals dedicate significant resources to the development of written communication to guide public health knowledge, attitudes, and behaviours [18]. This is often carried out with little regard for the social and political context in which these messages are developed and adopted, reducing their effectiveness [18]. The second strength of CDA is its ability to fill this gap by providing a methodology by which to analyze public beliefs about health, the construction of health in health promotion and mass media, and interactions between health professionals and patients. Not all research questions in PPH are best answered by ideologically driven methods. For example, a research question may ask, "Do school-based nutritional food provision programs result in decreased obesity among elementary school students, compared with school-based nutrition education"? This question may not require considerations of health ideology, and CDA may not be the ideal methodology to answer this question. In contrast, a research question may ask, "How does school staff knowledge and attitudes about the dietary behaviours of newcomer families affect the participation of newcomer children in school-based nutrition programs?" This question may require considerations of ideology, and a CDA approach may provide the best answer.

Finally, Evans-Agnew et al. (2016) describe how CDA is ideal for health policy research [3]. Specifically, it may be useful to examine the discourses that impede policy and those that promote it. Noting that most health policy research is conducted within a positivist research paradigm, the authors argue that CDA provides an alternate, more relevant research paradigm and strategy of inquiry for these purposes. This emphasis on alternative ways of knowing has increasingly been emphasized in the health sciences [3].

*5.2. Weaknesses*

One of CDA's most notable challenges articulates a key weakness: "CDA constantly sits on the fence between social research and political argumentation" ([6] p. 32). It is argued that the subjectivity of CDA is incompatible with the objectivity often sought in social scientific research. In response to this argument, CDA scholars assert that the social sciences are inherently subjective, making the pursuit of objectivity a futile endeavour. A researcher should not only acknowledge the subjective, but embrace its inevitability [19]. The second argument against this challenge comes from within the PPH community, specifically from critical public health scholars. It is argued that "the depoliticization of health serves powerful interests by delegitimizing analysis that might reveal and question

those interests" ([20] p. 122), concluding that the study of public health ought to be deliberately politicized. CDA inquiry strategies are not concerned with depoliticized objectivity, but with deliberate subjectivity, making it a useful tool for a range of PPH research inquiries.

The other prominent challenge to CDA is that its conclusions are rarely generalizable. Generalizability is dependent on the degree to which a research sample is representative of a population, allowing one to extend research findings outside the scope of the research project. This is of great importance in quantitative PPH research [11]. As a result of their commitment to social justice, CDA researchers should not be concerned with generalizability, but rather with identifying and challenging the structures that impact the research participants involved in the study. This requires contextualization, which may inevitably result in less generalizable results. However, CDA research findings may be transferable to other context and settings. Transferability is a concept in qualitative research that refers to the ability to apply qualitative research findings to other contexts and populations. Transferability may sound similar to generalizability, but the key difference is that the research sample is not required to be representative of a larger population and may or may not share certain qualities that allow for transferability. Moreover, whereas a lack of generalizability may be considered a limitation in quantitative research, a lack of transferability is not a drawback in qualitative research. Transferability is a desired aspect of qualitative research rather than a fundamental criterion for evaluating a study [19].

*5.3. Benefits to PPH*

The primary contribution of CDA to PPH research is its alignment with PPH's fundamental goals and alternate ways of seeking information. It has been argued that PPH research is frequently influenced by the research paradigms of the biomedical sciences. As a result, the wrong questions are often asked, the wrong methods are often used, and the wrong conclusions are often drawn to affect meaningful change in population health [3]. PPH research ought to fulfill key aims of PPH: to address the social determinants of health and reduce health inequities. This purpose is consistent with the principles of critical studies and social justice. This objective often requires researchers to focus on how individuals socially stratify, use their power, and construct health. CDA research accomplishes this through its critical impetus and constructivist orientation.

Additionally, while not all PPH research and initiatives focus specifically on marginalized populations, this is an important area of emphasis, because marginalization is a significant social determinant of health. The analysis of how language serves to produce and maintain uneven societal power relations [18] is fundamental to CDA; thus, it is a useful tool for examining how dominant discourses construct marginalized populations and health in a way that further impedes health for already marginalized communities.

## 6. Conclusions

This paper has explored the value of critical discourse analysis to population public health research. There are three principles that are crucial to the effectiveness and success of CDA: (1) CDA research should contribute to social justice; (2) CDA is strongly based in theory; and (3) CDA draws from constructivist epistemology. As with any other methodology, CDA has both strengths and weaknesses. Its strengths include that its critical impetus aligns with the social justice orientation of PPH, its attention to ideology enables effective inquiry into public beliefs about health, and its constructivist roots make it ideal for analyzing how health policy is formed and implemented. Weaknesses or challenges posed to CDA include that it is often excessively politicized, and its findings are often not generalizable. This analysis demonstrates the need for researchers in population public health to consider critical discourse analysis as an approach to understanding the social determinants of health and eliminating health inequities in order to achieve the health and wellness of all.

Through this exploration of the value of critical discourse analysis in public health research, the authors have learned and demonstrated the following. CDA is a method of examining how meaning about a particular phenomenon is constructed through language within a socio-political context. Intrinsic to CDA is the analysis of how language serves to produce and maintain societal power relations [8,9]; thus, it serves as a useful tool in examining how dominant discourses construct health issues. The aims of PPH, particularly the elimination of health inequity, is inherently socio-political. A PPH approach posits that health is socially, economically, and environmentally determined and that health inequities are the result of unfair inequalities in the distribution of social, economic, and environmental resources and benefits. The de-politicization of population health studies serves to reinforce the systems that produce health inequities [9]. The intrinsically political and critical stance of CDA allows us to challenge the socio-political structures and processes that create health inequities as a first step to eliminating them, fulfilling a core objective of population public health. The exploration in this study can serve as a step toward transforming how PPH research is approached in the future.

**Author Contributions:** Conceptualization, J.N., E.O.P. and T.C.T.; methodology, J.N., E.O.P. and T.C.T.; resources, E.O.P. and T.C.T.; data curation, J.N.; writing—original draft preparation, J.N.; writing—review and editing, E.O.P. and T.C.T.; supervision, E.O.P. and T.C.T. All authors have read and agreed to the published version of the manuscript.

**Funding:** This research received no external funding.

**Institutional Review Board Statement:** Not applicable.

**Informed Consent Statement:** Not applicable.

**Data Availability Statement:** Not applicable.

**Conflicts of Interest:** The authors declare no conflict of interest.

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
