# Peer review of "A Critical Lens on Health: Key Principles of Critical Discourse Analysis and Its Benefits to Anti-Racism in Population Public Health Research"

_societies, doi:10.3390/soc13020042_

Round 1
Reviewer 1 Report
This paper is a very interesting and orderly approach to CDA. I believe it is a valuable work that needs some improvement. In this sense, these comments, humbly, are intended to improve the work and help the authors. I hope they are understood in this way.
Section 3.1. begins in a very abrupt way. Before starting this section, it would be interesting to show the relationship between CDA and social justice. In addition, and on the other hand, it would be very important to go deeper into the explanation of what social justice is for the authors, since there are variations in the concept.
Section 3.2. is excessively brief and focuses on Wodak and Meyer's proposal, although Giddens is also mentioned. Undoubtedly, CDA is broader, as is well indicated in the article, and for this reason needs further development. For this reason, it is important to broaden the theoretical aspects, especially in point 3.2.1. when reference is made to social theories. In this sense I see, for example, a clear relationship with the texts of Isaac Ariail Reed "Interpretation and social knowledge" and also with the theoretical proposal of Bent Flyvbjerg in relation to phronesis in the social sciences. In turn, point 3.2.2. needs further development. Symbolic interactionism is a very important approach and should be given the importance it deserves.
The article should show concrete examples of the use of CDA and, in addition, show how this tool can be used concretely. The authors give a brief explanation of its application, but it would be useful to go into much more detail.
Point 4.1. is again too brief. On the basis of what has been said in this section, it is not clear how CDA differs from hermeneutics, phenomenology, social ethics, etc. Therefore, it is not clear to me whether a reader could understand the importance of CDA.
The references do not fit the template.
Author Response
This paper is a very interesting and orderly approach to CDA. I believe it is a valuable work that needs some improvement. In this sense, these comments, humbly, are intended to improve the work and help the authors. I hope they are understood in this way.
Thank you for reviewing and providing feedback on our manuscript. We have made the modifications outlined below in response to your suggestions.
Section 3.1. begins in a very abrupt way. Before starting this section, it would be interesting to show the relationship between CDA and social justice. In addition, and on the other hand, it would be very important to go deeper into the explanation of what social justice is for the authors, since there are variations in the concept.
Thank you for your feedback. We have added sentences to demonstrate concisely the connections between CDA and social justice.
131 – 135: “This principle is intertwined with the three aforementioned tenets of critical discourse analysis. CDA is inherently critical, concerned with ideology, and committed to expose the power dynamics underlying the phenomena it studies in order to eliminate disparities. It is therefore closely related to the aims of social justice to achieve a fair distribution of the benefits, responsibilities, and consequences of society.”
Also, we have added a section on the our interpretation of social justice.
136 – 145: “In the context of population public health, social justice is the view that everyone deserves equal rights and opportunities for good health.(8) This concept is closely re-lated to the concept of health equity, which is a core value of population public health. Health equity refers to social justice regarding health and the opportunity to attain health. Health inequities are avoidable and unfair disparities in health outcomes across populations. They are produced and reproduced by institutions, policies, and practices that create an unequal distribution of power and resources among communities based on race, class, gender, location, and other factors. Health inequities are social injustices in health. Consequently, the eradication of health inequity entails the eradication of social injustice in health.(9)”
Section 3.2. is excessively brief and focuses on Wodak and Meyer's proposal, although Giddens is also mentioned. Undoubtedly, CDA is broader, as is well indicated in the article, and for this reason needs further development. For this reason, it is important to broaden the theoretical aspects, especially in point 3.2.1. when reference is made to social theories. In this sense I see, for example, a clear relationship with the texts of Isaac Ariail Reed "Interpretation and social knowledge" and also with the theoretical proposal of Bent Flyvbjerg in relation to phronesis in the social sciences. In turn, point 3.2.2. needs further development. Symbolic interactionism is a very important approach and should be given the importance it deserves.
Thank you for these valuable insights. According to your suggestion, we felt it was essential to expand the section on symbolic interactionism for the purposes of this article. We have added the following content:
190 – 202: “Symbolic interactionism posits that a physical reality exists only through a person’s social understanding of that reality. Thus, when people act in relation to their surroundings, they do not do so directly, but rather indirectly through the lens of their social un-derstanding.(11,12) There are four main principles of SI. First, individuals act according to their social understanding of “objects” in their environment. For example, a person who views the “object” of the family as relatively unimportant will deemphasize the importance of family in their decisions and actions. Second, interactions occur in a social and cultural context in which objects, people, and situations must be defined and char-acterized based on an individual’s social understanding. Third, social understanding is created through interactions with other individuals and society. Fourth, these social un-derstandings are created and recreated through a process of interpretation that occurs each time a person interacts with others.(13)”
The article should show concrete examples of the use of CDA and, in addition, show how this tool can be used concretely. The authors give a brief explanation of its application, but it would be useful to go into much more detail.
Thank you for this feedback. We have added a section containing a concrete example of the use of CDA in public health.
- Critical Discourse Analysis of a Population Public Health Issue – Example
256 – 278: “An example of an existing critical discourse analysis of a PPH issue is done by Reitmanova, Gustafson, and Ahmed’s (2015) analysis of the Canadian Press and its im-plications for public health policies. Using framing as a theory of media effects, the authors conducted a critical discourse analysis of 273 articles from 10 major Canadian news sources over ten years. Framing aims to explain how news media cover, construct, and represent certain stories. This requires analyzing news reports for “the presence or absence of certain keywords, stock phrases, stereotyped images, sources of information and sentences that provide thematically reinforcing clusters of facts or judgments.”(17p3)
This analysis is valuable to PPH because the media influences public opinion and perception of health issues, as well as public health policy and healthcare practice. Thus, it is essential to understand how the media constructs and reports on health. More specific to this paper and to the goals of PPH, it is important to understand how the media constructs and portrays a population of Canadians who use the health system.
Reitmanova, Gustafson, and Ahmed’s (2015) found that Canadian news media discourses construct the immigrant body as both a disease breeder and an irresponsible fraudster. Moreover, these constructs are predicated on the racialization of immigrants and immigrant health issues. The results of this study suggest: 1) the de-racialization of immigrant bodies and immigrant health issues is required for more fair and accurate media coverage on immigrant health, and 2) the transformation of the Canadian press toward greater inclusiveness. These steps are needed to create the necessary shift for immigrants to receive equitable health care access.(17)”
Point 4.1. is again too brief. On the basis of what has been said in this section, it is not clear how CDA differs from hermeneutics, phenomenology, social ethics, etc. Therefore, it is not clear to me whether a reader could understand the importance of CDA.
Thank you for your feedback. We have added a few statements that explain why we believe CDA is ideally suited for PPH research and connected to our existing content.
284 – 292: “Although there may be other qualitative research methodologies with links to social jus-tice, CDA has been demonstrated to align with social justice and the core competencies of PPH. According to Edwards and Davison, PPH uses advocacy, policy change, and so-cial interventions to improve collective health; thus, social justice is a core value of PPH.(8) This is reflected in the Public Health Agency of Canada’s core competencies. The compe-tencies that align with social justice and CDA principles are shown in Table 2, adapted from Edwards and Davison (2008). In light of this, we contend that CDA is a critical qualitative method that is ideally suited for population public health research.”
6) The references do not fit the template.
Thank you. We have made changes to the font and size of text in order to fit the template.
Reviewer 2 Report
See attached

Author Response
Abstract Too general. I could not locate the main specific point of the article from the abstract. The abbreviation PPH is mentioned without any reference what it refers to (line 13)
Thank you for your feedback. We have added a sentence that aims to identify the purpose of the article and how we will meet that purpose.
“The purpose of this article is to demonstrate the benefits of CDA to Population Public Health (PPH) research. We will do this by…”
We have also included the full term “population public health” to clarify the abbreviation.
43-46 # 1: it is already well known that CDA is contributing to social justice. The three points raised by the researcher is taken for granted in CDA research. I do now see a need or any addition here.
47: Thank you for sharing your perspective. While it may be well-known that CDA contributes to social justice in social science and humanities research, its use in population public health research is still emerging, and thus this point may be less well-known. We have added a qualifier to clarify that our three principles pertain to the application of CDA in PPH.
“…CDA research, particularly in PPH:”
50 on Is there a need to give the history of CDA here in an article? This might be suitable for a book chapter.
Thank you for your insightful perspective. We believe that removing this section would compromise the overall coherence of this piece, so we have chosen to keep it.
80 on Same note as above, un-need history of CDA.
Thank you for your valuable perspective on this matter. We believe that removing this section would compromise the overall coherence of this piece, so we have chosen to keep it.
116 “As opposed to instructions or suggestions on how to … ???specific activ-116 ities like data collection and analysis,” incomplete thought.
118: Thank you for flagging this. We added a word – “conduct” - to complete this sentence.
“As opposed to instructions or suggestions on how to conduct specific activities like data collection and analysis, these are the principles and perspectives by which CDA re-searchers should work."
120 “there is a right and wrong way to think and act as a CDA researcher.” Explain how.
Thank you for your feedback. As the section that follows this paragraph is intended to explain how, we have included a transitional sentence.
123: “As demonstrated in the preceding discussion and as observed by many CDA scholars, there is no right or wrong way to conduct research in CDA; nonetheless, there is a right and wrong way to think and act as a CDA researcher. In the following section, we will explain how.”
152-184 Too general and theoretical. The specific connection to the point of the article is not clear.
Thank you for your feedback. We have added statements at beginning of each subsection to clarify the connection to the article’s main idea for the reader.
125 – 126: "The first essential principle for conducting effective CDA research in population health is that CDA research should advance social justice…"
162 – 163: "The second essential principle of effective CDA research in population public health is that CDA is theoretically grounded."
216 – 220: "The third essential principle of effective CDA in population public health research is that CDA draws from constructivist epistemology, which is vastly distinct from the prevalent positivist epistemological stance, frequently assumed in health science research. Constructivist research in PPH shares two characteristics: an explicit research paradigm and explicit reflexivity. We will elaborate upon these below."
This is an interesting piece of writing. However, it suits more to be a book chapter thank an article in a periodical.
Thank you for your thoughtful review and clear and useful feedback. We have made several changes outlined above in response to your suggestions.
Round 2
Reviewer 1 Report
The changes made allow for a better understanding of the research and its results. The importance of CDA for social justice in health is now clearer. I also think that an great effort to clarify microsociology has been made and I think the explanation in the article by Reitmanova et al. is very relevant. I thank the authors for their efforts. I believe the article has been improved and I am sure it will have the desired impact. Congratulations to the authors.
Author Response
Thank you su much for making time to review our manuscript. We totally appreciate your constructive comments which has helped us to improve the manuscript.